# Protein–Protein Interactions and Quantitative Phosphoproteomic Analysis Reveal Potential Mitochondrial Substrates of Protein Phosphatase 2A-B’ζ Holoenzyme

**DOI:** 10.3390/plants12132586

**Published:** 2023-07-07

**Authors:** Ahmed Elshobaky, Cathrine Lillo, Kristian Persson Hodén, Amr R. A. Kataya

**Affiliations:** 1Centre for Organelle Research, Faculty of Science and Technology, University of Stavanger, N-4036 Stavanger, Norway; dshobaky84@yahoo.com (A.E.);; 2Botany Department, Faculty of Science, Mansoura University, Mansoura 35516, Egypt; 3Department of Plant Biology, Uppsala BioCenter, Linnéan Center for Plant Biology, Swedish University of Agricultural Sciences, P.O. Box 7080, 75007 Uppsala, Sweden; 4Department of Biological Sciences, University of Calgary, Calgary, AB T2N 1N4, Canada

**Keywords:** energy metabolism, Krebs cycle, mitochondria, phosphatase, phosphoproteomics

## Abstract

Protein phosphatase 2A (PP2A) is a heterotrimeric conserved serine/threonine phosphatase complex that includes catalytic, scaffolding, and regulatory subunits. The 3 A subunits, 17 B subunits, and 5 C subunits that are encoded by the Arabidopsis genome allow 255 possible PP2A holoenzyme combinations. The regulatory subunits are crucial for substrate specificity and PP2A complex localization and are classified into the B, B’, and B” non-related families in land plants. In Arabidopsis, the close homologs B’η, B’θ, B’γ, and B’ζ are further classified into a subfamily of B’ called B’η. Previous studies have suggested that mitochondrial targeted PP2A subunits (B’ζ) play a role in energy metabolism and plant innate immunity. Potentially, the PP2A-B’ζ holoenzyme is involved in the regulation of the mitochondrial succinate/fumarate translocator, and it may affect the enzymes involved in energy metabolism. To investigate this hypothesis, the interactions between PP2A-B’ζ and the enzymes involved in the mitochondrial energy flow were investigated using bimolecular fluorescence complementation in tobacco and onion cells. Interactions were confirmed between the B’ζ subunit and the Krebs cycle proteins succinate/fumarate translocator (mSFC1), malate dehydrogenase (mMDH2), and aconitase (ACO3). Additional putative interacting candidates were deduced by comparing the enriched phosphoproteomes of wild type and B’ζ mutants: the mitochondrial regulator Arabidopsis pentatricopeptide repeat 6 (PPR6) and the two metabolic enzymes phosphoenolpyruvate carboxylase (PPC3) and phosphoenolpyruvate carboxykinase (PCK1). Overall, this study identifies potential PP2A substrates and highlights the role of PP2A in regulating energy metabolism in mitochondria.

## 1. Introduction

Protein post-translational modifications (PTMs) are covalent additions or modifications by chemical groups to a protein after its synthesis. PTMs add an additional level of regulation on top of gene expression and mRNA translation. Phosphorylation is a common PTM and a main regulator of cell signaling that is involved in a vast number of cellular processes [1]. Reversible phosphorylation is achieved through the balance between protein kinases, which act by adding a phosphate group to amino acids in the protein chain, and phosphatases, which remove phosphorylated groups from proteins [2]. An attestation to the importance of protein phosphorylation events, the *Arabidopsis thaliana* genome contains around 940 kinases and 150 phosphatases [3]. Protein phosphorylation and dephosphorylation processes regulate multiple downstream cellular processes (e.g., transcription, energy metabolism, and development) by affecting protein attributes such as localization, interactions, and activity [1,3].

A meta-analysis of phosphoproteomic data revealed that serine (Ser) is the most abundantly phosphorylated residue in plants (~85%), followed by threonine (Thr; ~15%) [4]. Consequently, one of the major phosphatase families in plants is composed of the serine/threonine-specific phosphoprotein phosphatases (PPPs). Among the PPPs, PP1 and PP2A are the largest groups in *Arabidopsis* [2,5]. Plant PP2A enzymes have been shown to be involved in various functions, such as the regulation of light signaling, hormone signaling, metabolism, other PTMs, and defense signaling [5,6]. PP2A enzymes are heterotrimeric proteins composed of a catalytic C subunit with regulatory (B) and scaffolding (A) subunits. The regulatory subunits are essential for substrate specificity and the localization of the complex. The complexity of PP2A enzymes is achieved through a combination of a low number of C subunits with A and B subunits [7]. Thus, despite having only five C subunits, *Arabidopsis* can theoretically attain over 250 PP2A variants (3 A subunits and 17 B subunits). The plant B family is further divided into the B, B’ and B’’ subfamilies, of which the B’ gene family contains nine members: α, β, γ, δ, ε, ζ, η, θ, and κ [2]. We here focus on the B’ subunit ζ (Z). 

The plant immune system needs tight negative regulation to avoid over-activation and autoimmunity. Protein kinases and phosphatases have been suggested as key players in plant immunity regulation, and some emergent studies have implicated PP2A in this role [8,9,10]. More specifically, all four members of the B’η subgroup (η, γ, θ, and ζ) were shown to be negative regulators of plant PAMP-triggered immunity. Increased resistance to the virulent bacterial strain *Pseudomonas syringae* DC3000 was observed in knockout mutants of *b’θ*, *b’ζ*, and *b’η*. Furthermore, the PP2A-B’ζ and PP2A-B’η complexes were found to control the activation of cell-surface pattern recognition receptors by regulating the phospho-status of the co-receptor BAK1 [8]. The *b’γ* mutant, on the other hand, displayed constitutive defense responses, and, in line with this, was implicated in resistance to a variety of infectious agents (aphids, hemibiotrophic bacteria, and necrotrophic fungi) [10,11].

Notably, it seems that the constitutive defense activation observed in the *b’γ* mutant can be explained by the involvement of the PP2A-B’γ complex in methionine recycling. By performing bimolecular fluorescence complementation (BiFC) experiments, Rahikainen et al. [12] first showed that B’γ interacts with indole glucosinolate methyl transferases. Further, B’γ was found to negatively regulate the metabolism of glucosinolates, which are important secondary metabolites that act as repellents against aphids and microbial plant pathogens in *Arabidopsis*. An increased build-up of specific compounds such as 4-methoxy-indol-3-yl-methyl glucosinolate is therefore a likely reason for the elevated defense responses of the *b’γ* knockout mutant. All of this solidifies the importance of the B’η subgroup in plant immune responses.

Mitochondria are double membrane-bound eukaryotic organelles. Reversible phosphorylation involving mitochondrial phosphatases has been largely neglected, although mitochondria are increasingly recognized as a hub for cell signaling, and several mitochondrial phosphatases have been reported to play vital functions. Nonetheless, evidence of the localization and activities of these reported phosphatases is lacking, and their functions and their mitochondrial translocation mechanisms are still poorly understood [13].

It has been reported that the PP2A regulatory subunit (B’ζ) targets mitochondria [7], and that when B’ζ mutates, plants not only show negative regulation of plant innate immunity, but they also develop a sugar-dependent phenotype [8,9]. This indicates a defect in energy metabolism in the *pp2a-b’ζ* mutant. Because seedlings of the knockout mutant of PP2A-B’θ, the nearest homolog of PP2a-B’ζ, have shown a defect in peroxisomal β-oxidation that is assumed to block the supply of succinate to the mitochondrial Krebs cycle during early seedling establishment [14], we investigated whether PP2A-B’ζ plays a putative role in the regulation of energy flow in mitochondria. We studied in vivo interactions with proteins involved in the energy flow to mitochondria and identified potential substrates of PP2A-B’ζ, employing protein–protein interactions and comparative quantitative phosphoproteomics. 

## 2. Methodology

### 2.1. Plant Material and Growth Conditions

Seeds of *Arabidopsis thaliana* wild type (Col-0) and two *pp2a-b’ζ* knockout lines (SALK_150586C (“*z1*”) and SALK_107944C (“*z2*”)) were surface-sterilized and stratified. The plants were grown under short-day or long-day conditions, as described in [15]. The mutant seeds were obtained from the European Arabidopsis Stock Centre (Nottingham, UK) and genotyped via PCR using T-DNA insertion primers obtained from the SALK institute’s SIGnAL website (http://www.signal.salk.edu/tdnaprimers.2.html), as is shown in Appendix A.

### 2.2. Gene Cloning for in Planta Expression In Vivo

The transient expression of constructs for BiFC was carried out using two systems: *Agrobacterium* infiltration into *Nicothiana benthamiana* leaves and gene gun delivery into onion epidermal cells. The BiFC constructs for the onion transformation were created by cloning the genes for the candidate interactors into pUC-VYNE_N173_ and by cloning the gene for B’ζ (At3g21650) into pUC-VYCE_C155_. For the *Agrobacterium* constructs, the plasmids hygII-VYNE_N173_ and kanII-VYCE_C155_ were used for the interactors and the B’ζ, respectively (Appendix A). For the *mMDH2* and *CSY5*, RNA was extracted from *Arabidopsis* flower tissue (Qiagen RNeasy Plant Mini, 74004), reverse transcribed using SuperScript IV (ThermoFisher, 18090010,Waltham, MA, USA), and PCR-amplified using Phusion polymerase (ThermoFisher, F531L). The genes from rest of the interactor candidates were obtained in plasmids from ABRC (https://abrc.osu.edu). The correctness of the inserts was verified via Sanger sequencing. All cloning primers are listed in Appendix A.

*N. benthamiana* plants were grown in soil under long-day conditions and transformed as described in [16]. The gene gun transformation into the onion epidermal cells was performed as in [15]. The interactions were analyzed two days post infiltration for the tobacco leaves and one to two days after bombardment for the onion cells, in accordance with [14]. Each interaction was analyzed in at least three independent experiments.

### 2.3. Sugar Dependence Assay

Seedlings were grown on a Linsmaier and Skoog (LS) medium for 7 days, both with and without sucrose (1%), under short-day conditions (light:dark 8 h:16 h) or in continuous darkness. The seedlings were scanned using a CANON scanner, and their hypocotyl length was measured using ImageJ (https://rsb.info.nih.gov/ij/).

### 2.4. Quantitative Phosphoproteomics

The total protein was extracted from 170 mg of tissue collected from six-day-old seedlings grown on a Murashige and Skoog (MS) medium without sugar (three samples each from the WT, *z1*, and *z2*), in accordance with the method described by Roitinger et al. [17]. The proteins were precipitated using the TCA/acetone method, collected via centrifugation, and processed for trypsin digestion using filter-aided sample preparation. Digestion was carried out overnight at 37 °C using 0.22 µg/µL MS-grade trypsin (Promega, PRV5111, Madison, WI, USA), and the samples were purified on peptide desalting spin columns (ThermoFisher 89851). The phosphopeptides were labeled with nine isobaric tandem mass tags using the TMT10plex kit (ThermoFisher, 90110) and enriched via sequential enrichment of metal oxide affinity chromatography (High-Select SMOAC; ThermoFisher, A32993). 

### 2.5. Liquid Chromatography-Tandem Mass Spectrometry (LC-MS/MS)

The tryptic peptides were analyzed via LC-MS/MS at the Southern Alberta Mass Spectrometry Facility using an Orbitrap Fusion Lumos Tribrid mass spectrometer (Thermo Fisher Scientific) controlled by Xcalibur (v4.0.21.10) and connected to a Thermo Scientific Easy-nLC 1200 system. Briefly, the Orbitrap first performed a full MS scan at a resolution of 120,000 FWHM to detect the precursor ions (375 *m*/*z* to 1575 *m*/*z*, a charge of +2 to +4). The AGC (auto gain control) and the max injection time were respectively set at 4 × 10^5^ and 50 ms. The most intense precursor ions that showed a peptidic isotopic profile and had an intensity threshold over 2 × 10^4^ were isolated using the quadrupole (window 0.7) and fragmented using HCD (38% collision energy) in the ion routing multipole. The fragmented ions (MS2) were analyzed at a resolution of 15,000, and the first mass was set to 100 in order to acquire the TMT reporter ions (AGC 1 × 10^5^ ms, max injection time 105 ms). To avoid the acquisition of the same precursor ion with a similar *m*/*z* (plus or minus 10 ppm), dynamic exclusion was enabled for 45 s.

### 2.6. Bioinformatics

Phosphopeptides that were more abundant in the two B’ζ mutants than in the WT were identified using the DEqMS method [18]. Data were normalized applying equal median normalization before applying DEqMS. Furthermore, unmodified peptides and peptides that had only been oxidized were removed from the dataset. In the comparison between the WT, *z1*, and *z2*, peptides were considered differentially phosphorylated if *p*_adj_ < 0.05 and absolute log_2_FC > 1. The proteins involved in energy metabolism were subjected to gene expression analysis using Genevestigator [19]. Mass spectrometry-verified phosphorylation sites were searched for in the candidate PP2A-B’ζ interactors using the PhosPhAt4 database (https://phosphat.uni-hohenheim.de/) [20]. Subcellular localization predictions were made using the tool TargetP 2.0 (https://services.healthtech.dtu.dk/service.php?TargetP-2.0) [21]. Enriched gene ontology (GO) terms [22,23] were identified in the combined phosphoproteome datasets from the WT and *z1*. The GO analysis was performed using topGO v. 2.50.0 [24] with the adjusted *p*-values from the DEqMS test performed on the WT vs. *z1* data. The parameters for the GO analysis test consisted of the default weight01 algorithm and Kolmogorov–Smirnov statistics. The annotations used were from the org. At.tair.db R package [25].

## 3. Results and Discussion

Our previous work showed that B’ζ mutant seedlings displayed a growth retardation phenotype on sucrose-free medium [9], indicating a defect in their energy metabolism. In addition, B’ζ was reported to be dually targeted to the cytosol and mitochondrion [7]. We hypothesize that the PP2A-B’ζ holoenzyme is involved in the regulation of the mitochondrial succinate/fumarate translocator, or that it affects the enzymes involved in the Krebs cycle. To test this, we set out to investigate the potential PP2A-B’ζ substrates via BiFC and phosphoproteomic analysis. 

### 3.1. Sugar Dependence Assay Points to a Role in Energy Metabolism

To further investigate the function of PP2A-B’ζ, we employed two homozygotic knockout lines, *z1* and *z2*. The knockout genotypes of both lines were confirmed using PCR. Line *z2* was previously observed to have impaired growth on a sucrose-free medium [9], whereas line *z1* was added to the present study to strengthen our phenotypic and functional observations. Both mutants displayed impaired hypocotyl elongation on sucrose-free media, especially when grown under continuous darkness (Figure 1), clearly indicating that the B’ζ subunit plays a role in energy metabolism. No phenotype was apparent in the *z1* or *z2* when they were grown on sucrose-containing MS media or in soil, a finding which agrees with the observations of Matre et al. [7].

These findings suggest that PP2A plays a putative role in the regulation of the energy metabolism through mitochondria. Therefore, we investigated the phosphor status of the enzymes of the Krebs cycle with the aim of identifying potential substrates for PP2A-B’ζ. The following seven candidates were targeted based on their documented key roles in the Krebs cycle: citrate synthase (CSY5), aconitase (ACO3), succinate dehydrogenase (SDH2-1), Mitochondrial succinate/fumarate carrier (mSFC1), succinyl-CoA-ligase (SCS), fumarase (FUM1), and mitochondrial malate dehydrogenase (mMDH2). Since all of the candidates except mSFC are part of multigene families in *Arabidopsis*, we carefully chose for each protein the paralog that was most relevant for our study. For example, ACO3 carries out the main aconitase activity of the three *Arabidopsis* ACOs [26], and FUM1 was preferred over FUM2 since it is mitochondrion-targeted and is essential (i.e., FUM2 cannot compensate for a lack of FUM1) [27].

### 3.2. Selected B’ζ Putative Interactors Carry Ser and Thr Phosphorylation Sites

First, we analyzed the gene expression pattern of *B’ζ* and the genes coding for the seven metabolic proteins using Genevestigator [19]. All genes were found to be expressed throughout development, but the *mSFC1*, *ACO3*, and *B’ζ* genes were found to be upregulated under leaf senescence (Appendix A). Next, we looked for evidence for Ser and Thr phosphorylation sites in the candidate substrates using the PhosPhAt4 database [20]. Experimentally verified phospho-Ser and phospho-Thr residues were identified in ACO3, FUM1, and SCS, making them plausible targets for dephosphorylation by PP2A-B’ζ (Appendix A). On the other hand, mSFC1, SDH2-1, mMDH2, and CSY5 each contain several predicted phosphorylation hotspots (Appendix A).

### 3.3. Verification of B’ζ Putative Interactors Involved in Energy Flow to Mitochondria

The genes encoding each enzyme and the carrier proteins were cloned, and the proteins were tested in vivo for their interactions with B’ζ using BiFC, a method that allows the detection of protein–protein interactions that under natural conditions are transient and thus hard to observe. Two heterologous systems for plant transient expression were used: tobacco leaves (agroinfiltration) and onion epidermal cells (biolistic bombardment). Since the mitochondrial localization of B’ζ depends on a free N-terminus [7], we tagged the B’ζ with part of the fluorescent protein Venus in its C-terminus. The other part of the fluorescent protein was linked to the C-terminus of each candidate interactor (Appendix A).

Through transient expression in tobacco leaves, the B’ζ fusion protein was observed to interact with mMDH2 in organelle-like structures which appeared to be the mitochondria (Figure 2a). This association of B’ζ with a key enzyme of the Krebs cycle provides a first link between PP2A and the mitochondrial energy flow. mMDH2 is one of two mitochondrial-localized MDH enzymes in *Arabidopsis*, and it catalyzes the conversion of malate into oxaloacetate in the Krebs cycle. The mMDH enzymes play key roles in the early life stages of *Arabidopsis*, as evidenced by the severe phenotype of the *mmdh1mmdh2* double knockout [28]. We also observed a positive interaction between the B’ζ and mSFC1 (Figure 2b). This association took place in the tobacco mesophyll cells, aligning with the reported localization of YFP-tagged B’ζ in *Arabidopsis* [7]. Additionally, a fluorescent signal was observed upon the investigation of the interaction between the B’ζ and ACO3 in the onion epidermal cells (Figure 3a). In addition, a physical interaction between the B’ζ and mSFC1 was confirmed (Figure 3c).

To determine the localization of the positive interactions, the mitochondrial marker pWEN95 was used. The mitochondrial marker harbors the mitochondrial import peptide from the β-subunit of the mitochondrial ATP synthase gene [29], which is fused with red fluorescent protein (RFP). The import peptide targets RFP to the mitochondrial matrix. The interaction complex, including the succinate/fumarate carrier mSFC1, was shown to be colocalized with the mitochondrial marker (Figure 3d), suggesting that the interaction of PP2A-B’ζ and mSFC1 takes place in the mitochondria or on the mitochondrial membrane in onion cells. Due to the mechanism of mitochondrial protein import, which requires protein unfolding for passage through the mitochondrial import system [30], the PP2A holoenzyme would need to disassemble before entry and then reform inside the mitochondrion. A mitochondrial targeting signal in one of the subunits would therefore not be able to direct the PP2A complex into the matrix. More likely, therefore, the interaction between mSFC1 and PP2A takes place on the mitochondrial surface. This interpretation agrees with the previously observed presence of the mSFC1-GFP fusion protein at the mitochondrial membrane in tobacco epidermal cells [31]. 

On the other hand, the interaction complex, including the ACO3, was seen to be cytoplasmic (Figure 3b). Overall, our BiFC data agree with the previously reported localization of B’ζ to both the mitochondrion and the cytoplasm [7]. ACO3 is an iron–sulfur-containing hydratase that is dually targeted to the mitochondrion and the cytoplasm in plants and yeast [26,32,33]. Mitochondrial ACO3 is active in the Krebs cycle, where it catalyzes the stereo-specific isomerization of citrate to isocitrate. The cytosolic form of *Arabidopsis* ACO3 has been implicated in the regulation of organellar reactive oxygen species (ROS) homeostasis and has been reported to bind to the PP2A-B’γ subunit [34]. Our observation of a cytosolic interaction between ACO3 and B’ζ indicates that PP2A-B’ζ could potentially be involved in ROS signalling as well, and that ACO3 could be putatively dephosphorylated by two different PP2A isozymes from the same PP2A subgroup.

The colocalization of B’ζ with mSFC1 and ACO3 is in line with concordant up-regulation of their three respective genes in senescent leaves (Appendix A). No interaction was seen between PP2A-B’ζ and CSY5, SDH2-1, SCS, or FUM1 in either the tobacco or the onion cells. It is possible that the dephosphorylation of these metabolic enzymes takes place during other developmental stages, in different tissue types, or under specific conditions not tested here. In summary, the BiFC revealed the cytoplasmic interactions between the PP2A-B’ζ and both the metabolic enzyme ACO3 and the transporter protein mSFC1. The latter interaction most likely takes place at the mitochondrial surface. All observations were strengthened by the occurrence of multiple independent transformation events and the use of two heterologous plant expression systems. 

The mSFC1 carrier protein plays an important role in the exchange of metabolites between organelles [35]. By inference from the homologous protein in yeast, *Arabidopsis* mSFC1 was first believed to import peroxisomal-produced succinate into the mitochondrion in exchange for fumarate, which in turn, via multiple downstream reactions, would be used in gluconeogenesis or in the glyoxylate cycle [27,35]. However, in the absence of biochemical evidence for this succinate/fumarase exchange by mSFC1, its function remained obscure until, more recently, it was revealed that mSFC1 transports citrate, isocitrate, and aconitate more efficiently than fumarate and succinate. Thus, it has been proposed that the main role of mSFC1 is the exchange of cytosolic citrate for mitochondrial isocitrate [31]. mSFC1 was also found to play an important role in seed germination and seedling development, and this is consistent with its expression in cotyledons, hypocotyls, and root tips [31,35]. Physical interaction between B’ζ and mSFC1 suggests that a putative dephosphorylation mediated by the PP2A-B’ζ complex could play a role in the regulation of metabolite exchange and cellular energy production in *Arabidopsis*.

### 3.4. Additional Metabolic Enzymes Revealed by Phosphoproteomics 

To further confirm the protein–protein interactions of PP2A-B’ζ with ACO3 and mSFC1, their encoding cDNAs were cloned into protein expression vectors and the in vitro interactions were investigated using microscale thermophoresis (MST). Recombinant proteins were produced in *Escherichia coli* and successfully purified, but the purified PP2A-B’ζ protein precipitated in every purified trial and hampered the planned MST analysis of the protein–protein interactions. We therefore decided to employ comparative quantitative phosphoproteomics as an alternative method for studying the B’ζ putative interactors. Quantitative phosphoproteomics is a method that can be used to identify and characterize PTM sites in proteins [36]. When performed on samples collected from suitable mutants, it can reveal novel substrates of kinases and phosphatases. Performing tandem mass tag (TMT) labeling prior to fragmentation increases the throughput by enabling the multiplex quantification of relative differences between the samples. In a typical phosphoproteomic workflow, proteins are first fragmented into peptides via trypsin digestion, then separated via liquid chromatography (LC), and finally quantified via mass spectrometry (MS). Phosphorylated sites are identified using bioinformatic methods applied to the MS data. In order to identify the substrates affected by the absence of B’ζ, all the proteins were extracted from the seedlings of the WT and the two knockout mutants, *z1* and *z2*. These were then digested in trypsin, and their phosphoproteomes were enriched using TiO_2_ so that they could be detected. The three phosphoproteomes were then compared. Although most phosphorylation events are transient, we expected to identify an accumulation of phosphorylated ACO3 and mSFC1, as well as other mitochondrial-related proteins, in the PP2A-B’ζ knockout mutants. 

A differential expression analysis (DEqMS) of the phosphoproteomic data was employed. Although neither the ACO3 nor the mSFC1 were differentiated in the mutants, the phosphorylated peptide form of the two mitochondrial proteins involved in plant energy metabolism (phosphoenolpyruvate carboxylase 3 (PPC3) and phosphoenolpyruvate carboxykinase 1 (PCK1)) were found differentially upregulated in one of the knockout mutants (Figure 4). These two enzymes, plus a third protein, pentatricopeptide repeat protein 6 (PPR6), had increased phosphorylation in the *z1* mutant compared with the WT (Appendix A). All of this, as well as the abolished PP2A-B’ζ function in the *pp2a-b’ζ* knockout mutant, suggests that PP2A-B’ζ plays a role in the mitochondrial metabolism and putatively increases possible PP2A-B’ζ substrates.

PPC3 catalyzes the carboxylation of phosphoenolpyruvate (PEP) into oxaloacetate (OAA), whereas PCK1 acts in the opposite direction (i.e., it forms PEP via OAA decarboxylation). PPC3 and PCK1 thereby play an important role in the regulation of the Krebs cycle. In fact, PCK1 serves as a cataplerotic enzyme, i.e., it removes intermediate metabolites from the Krebs cycle [37]. Interestingly, the two enzymes are known to be regulated by phosphorylation, and PP2A was found to be the phosphatase responsible for dephosphorylating of both PPC3 and PCK1 [38,39,40]. The identity of the regulatory subunit was, however, not discovered in these previous studies. The existence of Ser and Thr sites that are specifically more abundantly phosphorylated in the *z1* mutant is supported by MS data deposited in the PhosPhAt4 repository (Appendix A). The phosphorylation detected in the PPC3 (Ser 11) is also corroborated by a previously reported PP2A-medited dephosphorylation at an N-terminal Ser [41,42]. 

PPR proteins play important roles in the post-transcriptional regulation of mitochondrial genes [43,44]. They regulate many different aspects of the RNA life cycle, such as RNA stability, translation, processing, editing, and splicing. As such, PPR proteins are important for cellular respiration in plants. PPR6 is thought to contain a mitochondrial target peptide (TargetP 2.0, likelihood 0.9365), and therefore likely exerts its function inside the mitochondria. The maize homolog of PPR6 is responsible for the translation initiation and 5′ end maturation of the mitochondrial ribosomal protein S3 (*rps3*) mRNA, as well as the reduced translation and extended *rps3* mRNA 5′ end in the *mppr6* mutant [43]. Since *Arabidopsis PPR6* can complement the maize phenotype, the function of PPR6 seems to be conserved. We therefore hypothesize that PP2A-B’ζ, via dephosphorylation of the PPR6 protein, affects mitochondrial mRNA stability, with consequences for mitochondrial energy metabolism.

Taken together, we supply strong evidence of putative substrates of the PP2A-B’ζ holoenzyme in the mitochondria. ACO3 and mSFC1 were not found to be differentially phosphorylated in our comparison between the *z1*, *z2*, and WT phosphoproteomes. Potentially, PP2A-B’ζ shares these two substrates with the PP2A holoenzymes containing other regulatory B subunits, which would dephosphorylate ACO3 and mSFC1 in the absence of B’ζ. The difference between our BiFC observations and phosphoproteomic data could also be due to the different tissue types being used, i.e., leaves from adult tobacco plants and onion epidermal cells vs. six-day-old whole *Arabidopsis* seedlings.

To obtain additional insights into the pathways and processes that might be regulated by protein phosphorylation–dephosphorylation processes, a GO term enrichment analysis was carried out. Since the number of differentially phosphorylated proteins in the *z1* mutant was rather low, the combined phosphoproteomic datasets from the WT and *z1* were included in the pathway analysis. Notably, the terms phosphoenolpyruvate carboxylase activity (GO:0008964) and phosphoenolpyruvate carboxykinase activity (GO:0004612) were among the most significantly enriched terms in the molecular function domain (Appendix A); this supports the idea that reversible phosphorylation plays an important role in PEP ← → OAA conversion and the regulation of energy metabolism.

In summary, our phosphoproteomic analysis revealed for the first time that the PP2A-B’ζ subunit could dephosphorylate the two metabolic enzymes PPC3 and PCK1 in addition to the mitochondrial gene regulator PPR6. All of this could be linked to the role of the PP2A-B’ζ holoenzyme in plant innate immunity and energy metabolism, which are related to the reported phenotypes in their knockout mutants. 

## 4. Conclusions

The *PP2A-B’ζ* knockout mutants displayed impaired growth upon sucrose starvation, suggesting a deficiency in energy metabolism when the B’ζ regulatory subunit is removed. In addition, the TCA cycle enzymes PCK1 and PPC3, as well as the proposed mitochondrial gene expression regulator PPR6, were found to be differentially phosphorylated in the *pp2a-b’ζ1* mutant. Interactions between PP2A-B’ζ and the metabolic enzymes mMDH2 and ACO3 were confirmed in vivo. In addition, the mitochondrial metabolite carrier mSFC1 was shown to interact with B’ζ on the mitochondrial membrane. Taken together, the results of this work (Figure 5) suggest that the PP2A-B’ζ holoenzyme dephosphorylates several proteins involved in mitochondrial energy flow, and thus plays an important role in plant metabolism. 

## Figures and Tables

**Figure 1 plants-12-02586-f001:**
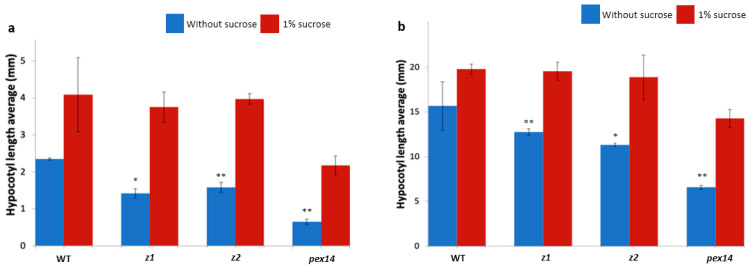
The *pp2a-b’ζ* knockout mutants showed reduced growth on media without sucrose. The hypocotyl lengths of the seedlings grown on MS media with or without sucrose in (**a**) 8 h light/16 h darkness or (**b**) continuous darkness were measured. The seedlings of the null mutant of peroxisomal PEX14 (“pex14”) were used as a control of the sucrose-dependent phenotype. Ten roots were measured per genotype, and the experiment was repeated four times. * *p* < 0.05 and ** *p* < 0.01; two-tailed *t*-test with equal variance (no sucrose vs. 1% sucrose). Error bars indicate standard deviations.

**Figure 2 plants-12-02586-f002:**
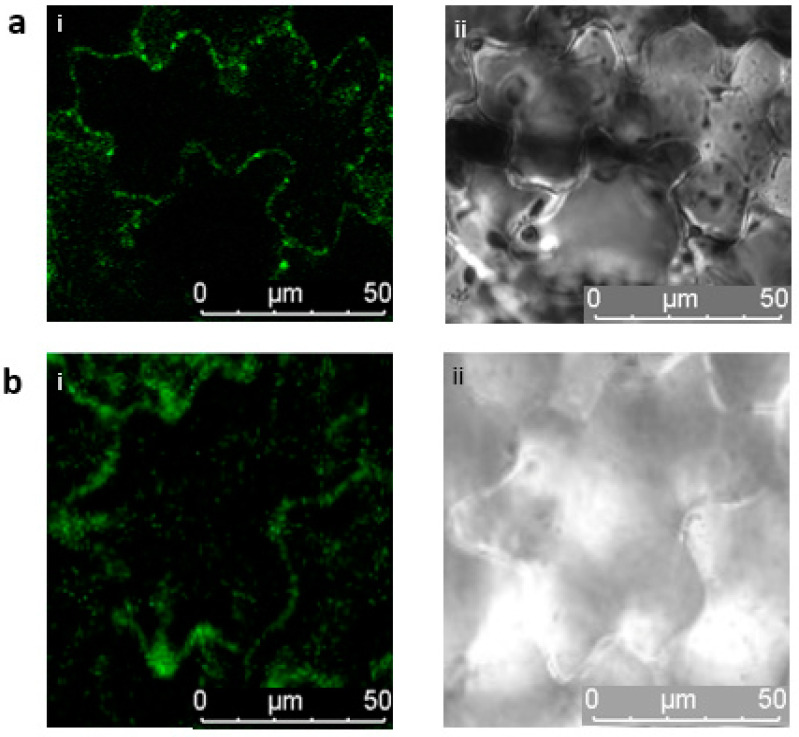
Interactions between PP2A-B’ζ and both mMDH2 and mSFC1. BiFC constructs were transiently expressed in tobacco leaf epidermal cells. The C-terminal part of Venus was fused to the B’ζ and the N-terminal Venus fragment was fused to (**a**) the mMDH2 and (**b**) the mSFC1. i: Venus; ii: brightfield. Both interactions are thought to be mitochondrial.

**Figure 3 plants-12-02586-f003:**
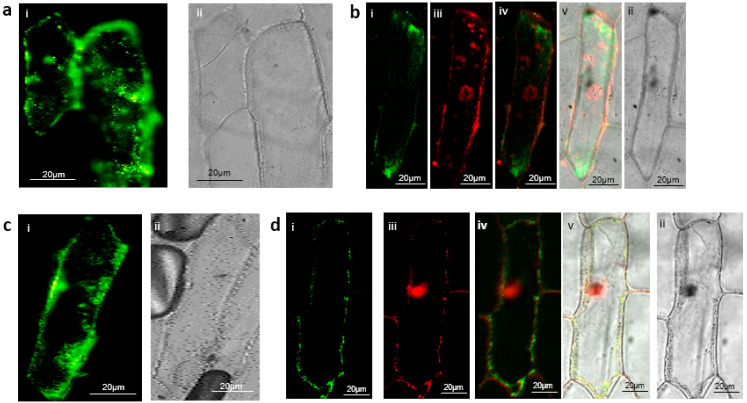
Visualization of PP2A-B’ζ interaction with mSFC1 and ACO3 in onion epidermal cells via BiFC. The B’ζ was terminally tagged with the C-terminal part of Venus while the N-terminal portion of Venus was fused to the C-terminus of (**a**,**b**) the ACO3 and (**c**,**d**) the mSFC1. i: Venus; ii: brightfield; iii: RFP (mitochondrial marker pWEN95); iv: overlay of Venus and RFP; v: overlay of Venus, RFP, and brightfield. In (**d**), the signals from Venus and the mitochondrial marker overlap, indicating the mitochondrial localization of the mSFC1-B’ζ interaction. The absence of overlap with the mitochondrial marker in (**b**) suggests the cytoplasmic localisation of the ACO3-B’ζ interaction.

**Figure 4 plants-12-02586-f004:**
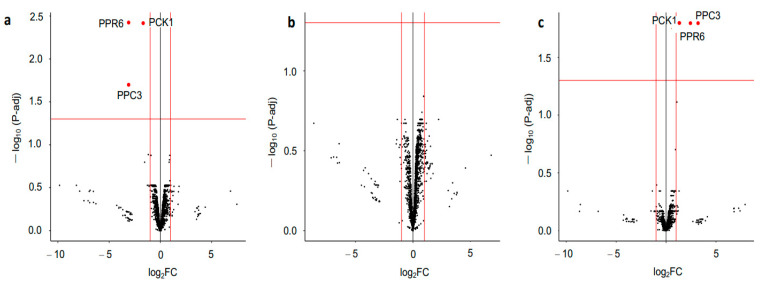
Volcano plots of quantitative phosphoproteomics. Phosphosites with fold changes >2 and adjusted *p* values < 0.05 were regarded as significantly differentially phosphorylated (indicated with red dots). (**a**): WT vs. *z1*. (**b**): WT vs. *z2*. (**c**): *z1* vs. *z2*.

**Figure 5 plants-12-02586-f005:**
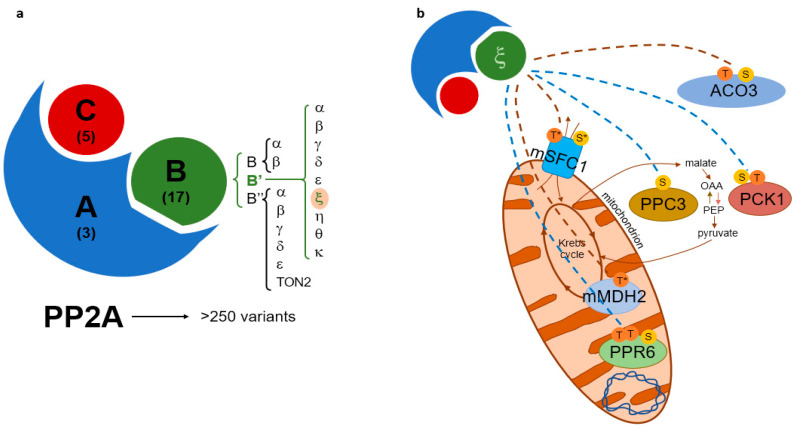
Models of PP2A subunit composition and proposed PP2A-B’ζ function. (**a**) The PP2A holoenzyme is composed of a scaffolding subunit (“A”), a regulatory subunit (“B”), and a catalytic subunit (“C”). The number of genes encoding each subunit in *Arabidopsis* is shown in parentheses. The focus of this study, ζ, belongs to the B’ protein family. (**b**) PP2A-B’ζ dephosphorylates multiple metabolic proteins. The BiFC-validated substrate interactions of B’ζ are indicated with brown dashed lines, and the B’ζ-dependent phosphorylation sites revealed via phosphoproteomics are indicated with blue dashed lines. The residues known to be phosphorylated are shown in the six substrates. S: serine; T: threonine; *: predicted phosphorylation. PCK1 catalyzes the conversion of oxaloacetate (OAA) into phosphoenolpyruvate (PEP), whereas PPC3 catalyzes the reverse reaction. Dephosphorylation of the mitochondrion-localized substrates could take place before entry into the mitochondrion or at the mitochondrial membrane (as is most likely for the carrier protein mSFC1).

## Data Availability

The unprocessed LC-MS/MS phosphosite data generated in this study are attached (Appendix A). The corresponding raw files are available upon request.

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
