# Peer review of "Protein–Protein Interactions and Quantitative Phosphoproteomic Analysis Reveal Potential Mitochondrial Substrates of Protein Phosphatase 2A-B’ζ Holoenzyme"

_plants, 2023, doi:10.3390/plants12132586_

Round 1
Reviewer 1 Report
The manuscript primarily validates three enzymes involved in mitochondrial energy metabolism that interact with PP2A-B'ζ through in vivo experiments. It identifies potential substrate candidates of PP2A through comparative phosphoproteomics data analysis between wild-type and PP2A-B'ζ knockout plants, providing a valuable addition to the field of PP2A research. The text is well-written, providing detailed background information, and the current results and conclusions are consistent, albeit somewhat limited. There are still many interesting aspects to explore regarding the model interpretation and underlying mechanisms.
1. For the potential PP2A-B'ζ substrates identified from the phosphoproteomics data, is there corresponding biFC data to support their interactions with PP2A-B'ζ?
2. For the three potential substrates identified from the biFC experiments, what could be the reasons behind the lack of significant changes in the phosphoproteomics data? Is there some literatures support 'Potentially, PP2A-B’ζ shares these two substrates with PP2A holoenzymes containing other regulatory B subunits, which would dephosphorylate ACO3 and mSFC1 in the absence of B’ζ'?
3. Please make an explanation of the red dot in the center of Figure3b(iii,iv) and Figure3d(iii,iv).
4. The index 'ii' in the figure legends of figure2 don't exist in the figure.
5. Conducting in vitro experiments to assess changes in protein enzymatic activity for all potential substrates would strengthen the arguments.
6. Are there any comparative data available for the levels of various metabolites in the Krebs cycle between wild-type and knockout plant strains?
Author Response
The manuscript primarily validates three enzymes involved in mitochondrial energy metabolism that interact with PP2A-B'ζ through in vivo experiments. It identifies potential substrate candidates of PP2A through comparative phosphoproteomics data analysis between wild-type and PP2A-B'ζ knockout plants, providing a valuable addition to the field of PP2A research. The text is well-written, providing detailed background information, and the current results and conclusions are consistent, albeit somewhat limited. There are still many interesting aspects to explore regarding the model interpretation and underlying mechanisms.
- For the potential PP2A-B'ζ substrates identified from the phosphoproteomics data, is there corresponding biFC data to support their interactions with PP2A-B'ζ?
Answer: Although we agree with the reviewer that a BifC experiment could add another confirmation of these potential substrates of PP2A-B'ζ, we did not perform these experiments as one of the candidates already was found to be dephosphorylated by PP2A not PP1 (Carter et al., 1990). Therefore, this clearly proves the importance of PP2A-B'ζ holoenzyme for these differentiated phopsho-substrates. We have highlighted this fact in the main text (lines 344-351) as follows:
“Interestingly, the two enzymes are known to be regulated by phosphorylation, and PP2A was found to be the phosphatase responsible for dephosphorylating both PPC3 and PCK1 [38,39,40]. The identity of the regulatory subunit was, however, not known in these previous studies. Meanwhile, the Ser and Thr sites that are specifically more abundantly phosphorylated in the z1 mutant are supported by MS data deposited in the PhosPhAt4 repository (Supplementary Table S2). The detected phosphorylation in PPC3 (Ser 11) is also in agreement with a previous reporting PP2A-medited dephosphoryla-tion at an N-terminal Ser [41,42].”
The main purpose of this study is to investigate the presence and find potential substrates of PP2A-B'ζ that are relevant to the PP2A-B'ζ function. Therefore, we have focused on BiFC interactions only for the candidates that were not shown in the phosphoproteomic data. We have successfully identified potential targets (phosphoproteomic- and BifC-based) of PP2A-B'ζ and supply findings that will support future focused studies investigating these interactions and help to reveal the relevance of these potential interactions and the functionality of phosphorylation of these substrates.
- For the three potential substrates identified from the biFC experiments, what could be the reasons behind the lack of significant changes in the phosphoproteomics data? Is there some literatures support 'Potentially, PP2A-B’ζ shares these two substrates with PP2A holoenzymes containing other regulatory B subunits, which would dephosphorylate ACO3 and mSFC1 in the absence of B’ζ'?
Answer: Phosphorylation is commonly considered to be a transient protein–protein interaction and finding of significant differentiated phosphoproteins usually is challenging. This fact could possibly affected the detetction of differentiated phosphorylation of the BifC-based interactions. However, in agreement with the reviewer, we cannot exclude a putative redundancy of other PP2A-holoenzymes and their ability to dephosphorylate these enzymes. We have highlighted this assumption in:
- (Lines 271-279): “ACO3 is an iron-sulfur-containing hydratase that is dually targeted to the mitochon-drion and the cytoplasm in plants and yeast [26,32,33]. Mitochondrial ACO3 is active in the Krebs cycle, where it catalyzes the stereo-specific isomerization of citrate to iso-citrate. The cytosolic form of Arabidopsis ACO3 was implicated in the regulation of or-ganellar reactive oxygen species (ROS) homeostasis and was reported to bind to the PP2A-B’γ subunit [34]. Our observation of cytosolic interaction between ACO3 and B’ζ indicates that PP2A-B’ζ could potentially be involved in ROS signalling as well, and that ACO3 could be putatively dephosphorylated by two different PP2A isozymes from the same PP2A subgroup.”
- (Lines 364-370): “ACO3 and mSFC1 were not found to be differentially phosphorylated in our comparison between the z1, z2 and WT phosphoproteomes. Potentially, PP2A-B’ζ shares these two substrates with PP2A holoenzymes containing other regulatory B subunits, which would dephosphorylate ACO3 and mSFC1 in the absence of B’ζ. The difference between our BiFC observations and phosphoproteomics data could also be due to the different tissue types being used – leaves from adult tobacco plants and onion epidermal cells vs. 6-day old whole Arabidopsis seedlings.”
- Please make an explanation of the red dot in the center of Figure3b (iii,iv) and Figure3d(iii,iv).
Answer: The red dot that is stained simultaneous to the mitochondrial targeting by the mitochondrial marker (Red fluorescence) refers to nucleus targeting. The nucleus identity can be seen in the bright field. Although PP2A-B’ζ has been also shown to target nucleus (Matre et al., 2009), the interaction of PP2A-B’ζ with these enzymes happened in the mitochondria and/or cytosol (Green Fluoresence), not in the nucleus.
- The index 'ii' in the figure legends of figure2 don't exist in the figure.
Answer: We have fixed the labels.
- Conducting in vitro experiments to assess changes in protein enzymatic activity for all potential substrates would strengthen the arguments.
Answer: The main target of our study is to decipher the presence of potential substrates of PP2A-B’ζ that will facilitate future studies to investigate the role of PP2A-B’ζ in mitochondria and energy metabolism. Our findings put us one step forward in this line of research, and we expect that several focused studies will benefit from our findings in the future. Therefore, by sharing our results with the scientific community we expect several studies to include experiments as recommended by the reviewer that will help to investigate the interaction conditions as well as if phosphorylation is indeed will affect the function of these enzymes. These experiments are expected to be challenging and are beyond the target of our study.
- Are there any comparative data available for the levels of various metabolites in the Krebs cycle between wild-type and knockout plant strains?
Answer: To our information, no comparative data between PP2A mutants and WT have been accomplished. Although, we agree with the reviewer that potential differentiation in Krebs cycle metabolites shall occur as we see an effect on the development of seedlings (Figure 1). This research was beyond our target of this study, although it will be interesting to see future investigations of these metabolites. The fact that our findings show potential interactions of Krebs-cycle related enzymes will further enhance this hypothesis, and will open doors to find the role of phosphorylation in the regulation of the function of the Krebs-cycle enzymes.
Reviewer 2 Report
lshobaky et al have identified the substrate proteins of the 3 phosphatase 2A-B’ζ holoenzyme by phosphoproteomic analysis. Interactions of the PP2A-B’ζ with Krebs cycle proteins Succinate/fumarate translocator (mSFC1), Malate dehydrogenase (mMDH2), and Aco-25 nitase (ACO3) have been confirmed. Other proteins, the mitochondrial regulator Arabidopsis pentatricopeptide repeat 6 (PPR6) and the two metabolic enzymes Phosphoenolpyruvate carboxylase (PPC3) and Phosphoenolpyruvate carboxykinase (PCK1) have been confirmed as putative targets/substrates of PP2A-B’ζ based on the evidence showing enriched phosphoproteomes of wild type and B’ζ mutants phenotypes. The manuscript is well-written and the results are described and interpretated appropriately. The manuscript has many strengths and can be accepted for the publication in Plants. I have the following concerns about manuscript.
Although the phosphorproteome analysis is clear and convincing, the measurement at such a crude level (total protein) brings some skepticism. The observed phosphorylation effect may not be the direct but can be the result of indirect cascade like mechanism. With this possibility, it would be in appropriate to claim that the said proteins are substrate of the PP2A-B’ζ.
The interaction between all said proteins with PP2A-B’ζ should be either confirmed and (if possible) qualified in vitro or in vivo at biochemical level. If this is not possible, then the suggested proteins can be claimed as something like “proteins being affected by the knock-out of PP2A-B’ζ”, not its substrate or interacting proteins.
Author Response
lshobaky et al have identified the substrate proteins of the 3 phosphatase 2A-B’ζ holoenzyme by phosphoproteomic analysis. Interactions of the PP2A-B’ζ with Krebs cycle proteins Succinate/fumarate translocator (mSFC1), Malate dehydrogenase (mMDH2), and Aco-25 nitase (ACO3) have been confirmed. Other proteins, the mitochondrial regulator Arabidopsis pentatricopeptide repeat 6 (PPR6) and the two metabolic enzymes Phosphoenolpyruvate carboxylase (PPC3) and Phosphoenolpyruvate carboxykinase (PCK1) have been confirmed as putative targets/substrates of PP2A-B’ζ based on the evidence showing enriched phosphoproteomes of wild type and B’ζ mutants phenotypes. The manuscript is well-written and the results are described and interpretated appropriately. The manuscript has many strengths and can be accepted for the publication in Plants. I have the following concerns about manuscript.
Although the phosphorproteome analysis is clear and convincing, the measurement at such a crude level (total protein) brings some skepticism. The observed phosphorylation effect may not be the direct but can be the result of indirect cascade like mechanism. With this possibility, it would be in appropriate to claim that the said proteins are substrate of the PP2A-B’ζ.
Answer: we wish to highlight that we agree with the reviewer that comparative phosphoproteomics shall be done by phospho-enriched peptides not from total proteins. In fact, we have performed our comparative phosphoproteomics in this study based on enriched phosphoproteomes not from the crude level (total proteins). We have isolated total proteins, digested to peptides, and enriched the phosphorylated peptides by TiO2 and quantitatively compared them based on isobaric (TMT) labeling tags. All of which gave us significant and trustable quantitative identification method.
In addition, the fact that one of the identified differentiated potential substrates has already been reported to be dephosphorylated by PP2A not PP1 (Carter et al., 1990), adds another line of evidence that our mutant missing PP2A-B’ζ is affecting the status of this substrate phosphorylation. We have highlighted this fact in the main text (lines 344-351) as follows:
“Interestingly, the two enzymes are known to be regulated by phosphorylation, and PP2A was found to be the phosphatase responsible for dephosphorylating both PPC3 and PCK1 [38,39,40]. The identity of the regulatory subunit was, however, not known in these previous studies. Meanwhile, the Ser and Thr sites that are specifically more abundantly phosphorylated in the z1 mutant are supported by MS data deposited in the PhosPhAt4 repository (Supplementary Table S2). The detected phosphorylation in PPC3 (Ser 11) is also in agreement with a previous reporting PP2A-medited dephosphoryla-tion at an N-terminal Ser [41,42].”
The interaction between all said proteins with PP2A-B’ζ should be either confirmed and (if possible) qualified in vitro or in vivo at biochemical level. If this is not possible, then the suggested proteins can be claimed as something like “proteins being affected by the knock-out of PP2A-B’ζ”, not its substrate or interacting proteins.
Answer: Bearing in mind that phosphorylation is commonly considered to be a transient protein–protein interaction, their interaction in-vitro is challenging. However, we have already expressed the proteins BiFC-based potential interactors and PP2A-B’ζ and attempted the interaction in vitro utilizing the MST method. However, PP2A-B’ζ quick precipitation hampered our ability to verify these interactions in vitro. We have referred to our trials in the lines (304-309):
“ To further confirm the protein-protein interaction of PP2A-B’ζ with ACO3 and mSFC1, their encoding cDNAs were cloned into protein expression vectors for inves-tigating the in vitro interaction employing microscale thermophoresis (MST). Recom-binant proteins were produced in Escherichia coli and successfully purified, but the pu-rified protein of PP2A-B’ζ precipitated in all purified trials and hampered the planned protein-protein interaction by MST analysis.”
But the fact that we identified ACO3 as interactor of PP2A-B’ζ by BiFC aligns with findings that the cytosolic ACO3 found interacting with the close homolog of PP2A-B’ζ, which is PP2A-B’γ. We have highlighted this fact as seen in the following lines (271-279):
“ACO3 is an iron-sulfur-containing hydratase that is dually targeted to the mitochon-drion and the cytoplasm in plants and yeast [26,32,33]. Mitochondrial ACO3 is active in the Krebs cycle, where it catalyzes the stereo-specific isomerization of citrate to iso-citrate. The cytosolic form of Arabidopsis ACO3 was implicated in the regulation of or-ganellar reactive oxygen species (ROS) homeostasis and was reported to bind to the PP2A-B’γ subunit [34]. Our observation of cytosolic interaction between ACO3 and B’ζ indicates that PP2A-B’ζ could potentially be involved in ROS signalling as well, and that ACO3 could be putatively dephosphorylated by two different PP2A isozymes from the same PP2A subgroup.”
Our BiFC- and phosphoproteomic-based findings align with the literature evidences of PP2A dephosphorylation of two of these enzymes, and the mutant phenotype. However, we expect future focused studies to investigate the occurrence and the importance of these interactions that may affect the Krebs cycle, and role of PP2A-B’ζ in regulating mitochondrial function during energy metabolism and immunity. Therefore, we have maintained labelling these substrates as potential, as the reviewer also suggests. Referring to these substrates as potential as seen in the title, will highlight the need to confirm these interactions under specific conditions/or in specific tissues as well as will highlight the crucial needs to identify the role of phosphorylation of these enzymes.
Reviewer 3 Report
In this article, the authors employed bimolecular fluorescence complementation in tobacco and onion cells to confirm the involvement of PP2A-B'ζ holoenzyme in energy flow to mitochondria. Subsequently, they utilized phosphoproteomics to identify potential interacting candidates, namely PPR6, PPC3, and PCK1.
The study is well-designed, and the chosen experimental approach is appropriate.
Suggestions:
#1. The authors utilized phosphoproteomics with the MOAC platform. Have the authors also performed total proteomics? It is recommended to compare phosphorylated proteins and protein abundance simultaneously, so the comparison can be more meaningful.
#2. Line 183: The mention of knockout lines is unnecessary since z1 and z2 have already been defined in Line 102.
#3. Figure 1: The presence of pex14 is not explained in the main text or figure legend. Please provide a clear explanation for its inclusion here.
#4. Since both z1 and z2 are mutants, and they exhibit different phosphoproteomics results as shown in Figure 4, the conclusion regarding PP2A-B'ζ holoenzyme dephosphorylating PPC3 and PCK1 should be verified, perhaps using western blot.
#5. The labels used in the manuscript are not consistent. Please use either "WT" or "Col-0" for consistency.
#6. The font size in the figures should meet the publisher's requirements.
#7. AGI identifiers are not recommended in Figure 4. Please replace them with protein names.
Author Response
In this article, the authors employed bimolecular fluorescence complementation in tobacco and onion cells to confirm the involvement of PP2A-B'ζ holoenzyme in energy flow to mitochondria. Subsequently, they utilized phosphoproteomics to identify potential interacting candidates, namely PPR6, PPC3, and PCK1.
The study is well-designed, and the chosen experimental approach is appropriate.
Suggestions:
#1. The authors utilized phosphoproteomics with the MOAC platform. Have the authors also performed total proteomics? It is recommended to compare phosphorylated proteins and protein abundance simultaneously, so the comparison can be more meaningful.
Answer: Our target in this study to identify substrates of PP2A that are related to putative role of PP2A-B’ζ in mitochondria. We have proved interactions in-vivo and attempted to identify affected proteins by the absence of PP2A-B’ζ in the knockout mutant, from which we have identified three mitochondria-related enzymes. The fact that we have identified three differentiated phosphorylated proteins, from which one has been already reported to be dephosphorylated by PP2A has given evidence that the variations are due to the lack of PP2A-B’ζ. We have highlighted this fact in the main text (lines 344-351) as follows:
“Interestingly, the two enzymes are known to be regulated by phosphorylation, and PP2A was found to be the phosphatase responsible for dephosphorylating both PPC3 and PCK1 [38,39,40]. The identity of the regulatory subunit was, however, not known in these previous studies. Meanwhile, the Ser and Thr sites that are specifically more abundantly phosphorylated in the z1 mutant are supported by MS data deposited in the PhosPhAt4 repository (Supplementary Table S2). The detected phosphorylation in PPC3 (Ser 11) is also in agreement with a previous reporting PP2A-medited dephosphoryla-tion at an N-terminal Ser [41,42].”
As referred to in the text and the title, we refer to these substrates as “potential” substrates of PP2A and future studies will reveal the nature of these interactions. Therefore, we did not attempt to compare the total proteome and we expect that our findings will be assessed in future focused studies on individual enzyme-based studies.
#2. Line 183: The mention of knockout lines is unnecessary since z1 and z2 have already been defined in Line 102.
Answer: These have been deleted.
#3. Figure 1: The presence of pex14 is not explained in the main text or figure legend. Please provide a clear explanation for its inclusion here.
Answer: the explanation was added in the figure legend as follows: “The seedlings of the null mutant of peroxisomal PEX14 “pex14” were used as a control of the sucrose dependence phenotype.”
#4. Since both z1 and z2 are mutants, and they exhibit different phosphoproteomics results as shown in Figure 4, the conclusion regarding PP2A-B'ζ holoenzyme dephosphorylating PPC3 and PCK1 should be verified, perhaps using western blot.
Answer: The fact that no specific phosho-antibodies against these targets are available makes verification of these possible substrates by western challenging and beyond our target from this study. We aimed to find potential substrates of PP2A-B'ζ that will help future studies to dissect the regulatory role of PP2A on mitochondrial functions with focus on energy metabolism. We successfully provide evidence that these potential substrates of PP2A-B'ζ aligns with the reported fact that PP2A have been proven to dephosphorylayte one of them. All of which, strengthen our findings in the mutant and open doors for further investigations in this field. This was referred in the text as the following:
We have highlighted this fact in the main text (lines 344-351) as follows:
“Interestingly, the two enzymes are known to be regulated by phosphorylation, and PP2A was found to be the phosphatase responsible for dephosphorylating both PPC3 and PCK1 [38,39,40]. The identity of the regulatory subunit was, however, not known in these previous studies. Meanwhile, the Ser and Thr sites that are specifically more abundantly phosphorylated in the z1 mutant are supported by MS data deposited in the PhosPhAt4 repository (Supplementary Table S2). The detected phosphorylation in PPC3 (Ser 11) is also in agreement with a previous reporting PP2A-medited dephosphoryla-tion at an N-terminal Ser [41,42].”
#5. The labels used in the manuscript are not consistent. Please use either "WT" or "Col-0" for consistency.
Answer: we have adjusted this.
#6. The font size in the figures should meet the publisher's requirements.
Answer: we have adjusted this.
#7. AGI identifiers are not recommended in Figure 4. Please replace them with protein names.
Answer: we have added protein names on the figures.
Round 2
Reviewer 2 Report
No further comments.